# Occurrence and Distribution of Strains of *Saccharomyces cerevisiae* in China Seas

**Bai-Chuan Tian [1], Guang-Lei Liu [1,2] , Zhe Chi [1] , Zhong Hu [3] and Zhen-Ming Chi [1,2,\*]**

1   College of Marine Life Sciences, Ocean University of China, Yushan Road, No. 5, Qingdao 266003, China;
    m15634129717@163.com (B.-C.T.); liugl@ouc.edu.cn (G.-L.L.); cz1108@ouc.edu.cn (Z.C.)
2   Laboratory for Marine Biology and Biotechnology, Qingdao National Laboratory for Marine Science
    and Technology, Qingdao 266003, China
3   Department of Biology, Shantou University, Shantou 515063, China; hzh@stu.edu.cn
\*   Correspondence: chi@ouc.edu.cn

**Abstract:** The yeast *Saccharomyces cerevisiae* has been widely applied in fermentation industries, chemical industries and biological research and it is widespread in different environments, especially in sugar-rich environments. However, little is known about the occurrence, distribution and roles of *S. cerevisiae* in marine environments. In this study, only 10 strains among all the yeasts isolated from different marine environments belonged to *S. cerevisiae*. It was found that most of the strains of *S. cerevisiae* in marine environments occurred in guts, the surface of marine fish and mangrove trees. In contrast, they were not found in seawater and sediments. All the strains of *S. cerevisiae* isolated from the marine environments had a lower ability to produce ethanol than the highly alcohol-producing yeast *Saccharomyces* sp. W0 isolated from fermented rice, but the strains 2E00400, 2E00558, 2E00498, 2E00723, 2E00724 could produce higher concentrations of ethanol than any other marine-derived strains of *S. cerevisiae* obtained in this study. However, some of them had higher ethanol tolerance and higher trehalose content than *Saccharomyces* sp. W0. In particular, ethanol tolerance of the yeast strain 2E00498 was higher than that of *Saccharomyces* sp. W0. This may be related to the harsh marine environments from which they were isolated. Such yeast strains with higher alcohol tolerance could be used to further improve the alcohol tolerance of *Saccharomyces* sp. W0.

**Keywords:** marine environments; *S. cerevisae*; identification; ethanol production; alcohol endurance

## 1. Introduction

Just like its name, *S. cerevisiae* is a functionally heterotrophic colonizer of sugary substrates [1]. It has also been found that it plays an important role in the production of natural secondary metabolites, pectinase and glycosidase, the inhibitory effect on the growth of mycotoxin-producing fungi, the degradation of some fractions of kasein and $CO_2$ evolution and has lipolytic, proteolytic and urease activities [2]. *S. cerevisiae* has been extensively applied in fermentation, food, chemical and pharmaceutical industries for a long time [2]. In order to produce a higher yield of ethanol and any other metabolites, it is important to screen and utilize highly alcohol-tolerant strains of *S. cerevisiae* from different environments, including marine environments [2].

It has been well documented that the oceans cover 71% of the surface of the Earth, and they contain abundant biotic resources, including marine yeasts, their functional genes, their metabolites and highly active enzymes [3]. In our previous investigations [4], strains of several yeast genera isolated from different marine environments included *Rhodotorula* spp., *Rhodosporidium* spp., *Candida* spp., *Debaryomyces* spp., *Cryptococcus* spp., *Yarrowia lipolytica*, *Aureobasidium* spp., *Metschnikowia* spp., *Torulopsis* spp., *Pichia* spp., *Kluyveromyces* spp., *S. cereviaise*, *Williopsis* spp., *Pseudozyma* spp., *Hansenula* spp., *Trichosporon* spp., *Filobasidium* spp., *Leucosporidium* spp. and many unidentified genera of yeast strains. Some psychrophilic yeasts such as *Mrakia frigida*, *Guehomyces pullulans* and *Metschnikowia australis*

have also been found to be widely distributed in the sea sediment of Antarctica [5,6]. All these yeasts were obtained from samples of seawater, sediments, marine fish, marine algae and mangroves in the Yellow Sea, East China Sea, South China Sea, Pacific Ocean, Indian Ocean, Antarctica and deep sea. Nagahama [7] reported that there were around 30 yeast species isolated from sediment and invertebrates collected from deep-sea floors around the northwest Pacific Ocean. However, *Saccharomyces* spp. were not found in such deep-sea floors. It was also stated in his review article that *S. cerevisiae* did not occur in offshore, estuary and mangrove areas. In contrast, *S. cerevisiae* can be found in the gastrointestinal tract of healthy Oncorhynchus mykiss and colonize the intestine of rainbow trout and sea bass [8,9]. This may suggest that *S. cerevisiae* only grow and survive in environments rich in organic substances. Harsh marine environments are characterized by a lack of organic nutrients, low temperature, high pressure and high concentration of salts [3]. We think that such harsh environments may make *S. cerevisiae* develop some special properties, such as tolerance to stressful conditions [4]. Therefore, it may be possible to obtain strains of *S. cerevisiae* with high alcohol tolerance from marine environments. Thus, the main aims of this study are to survey the occurrence and distribution of *S. cerevisiae* in different marine environments in China in order to select the strains of *S. cerevisiae* with high alcohol tolerance for their application in alcohol industries and basic research in biotechnology.

## 2. Materials and Methods

### 2.1. Yeast Strains

*Saccharomyces* sp. W0, which is both a highly alcohol-producing and highly alcohol-tolerant yeast strain, which was isolated from fermented rice and has been utilized for high ethanol production in our laboratory for over 20 years [10], was used as a representative of terrestrial yeasts in this study. Other strains of *S. cerevisiae* used in this study were isolated from different marine environments in China, as described below.

### 2.2. Medium and Chemicals

The medium for growth of the yeasts was a yeast–peptone–dextrose (YPD) medium (prepared with seawater or distilled water) containing 20.0 g $L^{-1}$ glucose, 20.0 g $L^{-1}$ peptone, 10.0 g $L^{-1}$ yeast extract. The alcohol fermentation medium was a synthetic medium [11] supplemented with 200.0 g $L^{-1}$ sucrose and 10.0 g $L^{-1}$ ammonium sulfate. All the chemicals used in this study were purchased from Sinopharm Chemical Reagents Co. Ltd., Shanghai, China.

### 2.3. Sampling

Hypersaline seawater (1 m depth, 26 °C, pH 8.1 and 15% salinity, March of 2004), sediments of the salterns (1 m depth, 26 °C, pH 8.1 and 15% salinity, October of 2004), seawater, sediments, different species of marine animals and algae in the Yellow Sea, Bohai Sea, China East Sea and South China Sea (10 m depth, 15 °C, pH 8.1 and 2.89% salinity, October of 2005) were collected. The roots, stems, branches, leaves, barks, fruits and flowers (35 °C, July of 2008) from 18 species of mangrove plants at 6 different places in Hainan, Guangdong and Fujian Provinces in China were also used as sources for yeast isolation in this study. At the same time, the roots, stems, branches, leaves, barks, fruits and flowers (35 °C, July of 2008) from *Clerodendrum inerme*, *Cassytha filiformis* L., *Phragmites communis* Trin., *Pandanus tectorius* and *Pluchea indica* Less that are the typical accompanying plant species distributed in the mangrove ecosystems mentioned above were also utilized as sources for yeast isolation in this study. The seawater and sediments (35 °C, July of 2008) in some of the mangrove ecosystems mentioned above were also collected for isolation of the yeasts.

### 2.4. Isolation and Purification of Marine Yeasts

First, all the marine animals, marine algae and the roots, stems, branches, leaves, barks, fruits and flowers from the different mangrove trees were surface disinfected using

70% (*v*/*v*) ethanol. Then, after collection and treatment of the samples, two milliliters of the seawater or 2.0 g of the sediments or 2.0 g of the disinfected roots, stems, branches, leaves, barks, fruits and flowers from different mangrove trees or 2.0 g of the skin, gills and contents of the gut from different marine animals or 2.0 g of the different marine algae were immediately suspended in 50.0 mL of the sterile YPD medium supplemented with 5.0 g L$^{-1}$ chloramphenicol in a 250 mL shaking flask, respectively. Finally, all the flasks were aerobically cultivated at 25 °C and 180 rpm for five days. After suitable dilution of the cell cultures with sterile saline water, each dilute was evenly spread on a YPD plate with 5.0 g L$^{-1}$ chloramphenicol and all the plates were incubated at 20–25 °C for five days. Each colony from the plates was and purified and transferred to a YPD slant for further cultivation, respectively.

### 2.5. Identification of the Yeasts

Each colony obtained above was photographed and recorded. The cell morphology of each yeast strain on the YPD plate was observed and photographed under a microscope. The fermentation and assimilation tests for each yeast strain were performed using the methods described by Kurtzman and Fell [12]. At the same time, *S. cerevisiae* ATCC32703 was used as a type strain.

### 2.6. DNA Extraction and PCR Reactions

Each yeast strain obtained above was cultivated in the liquid YPD medium at 28 °C and 180 rpm for 20 h. The total genomic DNA of each yeast strain was isolated and purified by using the methods described by Sambrook et al. [13]. Amplification and sequencing of the 26S rDNA sequence from the genomic DNA of each yeast strain were performed according to the methods described by Sambrook et al. [13], Chi et al. [14] and Gao et al. [15]. The common primers for amplification of the 26S rDNA from the DNA template of each yeast strain were used, the forward primer was NL-1 (5′-GCATATCAATAAGCGGAGGAAAAG-3′) and the reverse primer was NL-4 (5′-GGTCCGTGTTTCAAGACGG-3′) [16]. The reaction system (25.0 μL) was composed of 10 × buffer 2.5 μL, dNTP 0.8 μmol L$^{-1}$, MgCl$_2$ 1.5 mmol L$^{-1}$, NL-1 0.5 μmol L$^{-1}$, NL-4 0.5 μmol L$^{-1}$, Taq DNA polymerase 1.25 U, template DNA 1.0 μL and H$_2$O 16.6 μL. The conditions for the PCR amplification were as follows: initial denaturation at 94 °C for 10 min, denaturation at 94 °C for 1 min, annealing temperature at 53 °C for 1 min, extension at 72 °C for 2 min, final extension at 72 °C for 10 min. PCR was run for 30 cycles and the PCR cycler was the Mastercycler gradient thermal cycler made by Eppendorf. The obtained PCR products were separated by agarose gel electrophoresis and purified by using TAKARA DNA gel purification kits (TAKARA, Japan). The purified PCR products from each DNA template were ligated into the pMD-19T vector and the recombinant plasmid was transformed into the competent cells of *Escherichia coli* DH5α. The transformants obtained were selected on LB plates supplemented with ampicillin. The plasmids in the transformant cells were extracted by using the TIANGEN plasmids extraction kits (TIANGEN, Beijing, China). The 26S rDNA fragments inserted in the vector were sequenced by Shanghai Sangon.

### 2.7. Phylogenetic Analysis

The 26S rDNA sequences obtained above were aligned by using BLAST analysis (http://www.ncbi.nlm.nih.gov/BLAST, 20 May 2010). The sequences which shared over 98% similarity with the currently available 26S rDNA sequences of fungi were considered to be the same species and multiple alignments were performed by using ClustalX 1.83 and the phylogenetic tree was constructed by neighbor-joining using MEGA 4.0 [17].

### 2.8. Alcohol Fermentation Tests

Each yeast strain of *S. cerevisiae* obtained above was grown aerobically at 30 °C and 180 rpm in synthetic medium [11] with 20.0 g L$^{-1}$ sucrose and 10.0 g L$^{-1}$ ammonium sulfate for 18 h. Around $3 \times 10^8$ cells mL$^{-1}$ in 15.0 mL of the fresh culture were collected by

centrifugation at $4000\times g$ and 4 °C for 5 min. The collected cells were transferred to a 300 mL bottle with 150.0 mL of the synthetic medium containing 200.0 g $L^{-1}$ sucrose and 10.0 g $L^{-1}$ ammonium sulfate and the final concentration of the yeast cells was $3 \times 10^7$ cells $mL^{-1}$ of the fermentation medium. Each bottle was fitted with a rubber bung perforated by a needle and incubated statically at 30 °C. The loss of weight by $CO_2$ liberation during the fermentation was monitored each day until the fermentation ceased. As the fermentation proceeded, additional sucrose (around 5.0 g per 100 mL of the fermentation medium) was added to keep a suitable concentration of sucrose in the medium. The final alcohol concentration in the fermented media was analyzed as described below.

### 2.9. Ethanol Shock Treatment

Each yeast strain of *S. cerevisiae* obtained above was inoculated into the YPD medium and cultivated at 30 °C and 180 rpm for 18 h as described above. The yeast cells from the culture ($1 \times 10^8$ cells $mL^{-1}$) were harvested by centrifugation and washed three times with sterile saline water. The washed cells were resuspended in 8.2 mL of sterile saline water and 1.8 mL of absolute ethanol was added to the cell suspension. Then, the cell suspensions were mildly shaken at 50 rpm in a water bath (30 °C). During the high ethanol shock treatment, the treated samples were taken periodically from the cell suspension and the cell viabilities were determined following appropriate dilutions of the cultures and plating the dilute on YPD plates. Each strain on the plates was grown at 30 °C for 48 h or 72 h and the colonies appeared on each plate were counted after 48 or 72 h incubation at 30 °C. All the plates were run in duplicate, with the results averaged for each duplication. Alcohol tolerance expressed as the percentage of the yeast survivors was determined by comparing the colony count of the ethanol-shocked cells with that of non-shocked controls [10]. The non-treated cells of the same strain were used as the control.

### 2.10. Ethanol Assay

One hundred milliliters of the fermented culture obtained above and 100.0 mL of distilled water were well mixed together. The mixture was distilled at 100 °C and 100 mL of the distillate was harvested. The distillate was diluted 1000-fold. The ethanol concentrations in the dilute and the standard ethanol solution were measured by using gas chromatography (HP5890II, Hewlett-Packard, USA). The chromatography column was a fused silica AC-20 capillary column (30 m × 0.25 mm i.d., 0.25 m film thickness). The GC conditions used were previously described by Wang et al. [18].

### 2.11. Trehalose Extraction and Assay

Each yeast strain obtained above was inoculated into the liquid YPD medium and cultivated at 30 °C for 18 h by shaking at 180 rpm. All the yeast cells in 5.0 mL of the culture were collected and washed by centrifugation at $5000\times g$ and 4 °C. All the trehalose in the washed yeast cells of each yeast strain was extracted three times with 0.5 M of trichloroacetic acid [19] and the amount of the extracted trehalose was assayed by using the anthrone method [19]. Finally, trehalose content per 100 g of cell dry weight was calculated based on the cell dry weight in 5.0 mL of the culture and was determined as described below.

### 2.12. Measurement of Cell Dry Weight

The yeast cells from 5.0 mL of the culture of each yeast strain were harvested and washed three times with sterile saline water by centrifugation at $4000\times g$ for 5 min. Then, all the washed yeast cells in each centrifuge tube were dried at 100 °C until the cell dry weight was constant.

### 2.13. Statistical Analysis

The data obtained above were subjected to one-way analysis of variance (ANOVA) [20]. P values were calculated by a Student's *t*-test (*n* = 3). *p* values less than 0.05 were considered

statistically significant. Statistical analysis was performed using a SPSS 11.5 for Windows (SPSS Inc., Chicago, IL, USA).

## 3. Results

### 3.1. Occurrence and Distribution of S. cerevisiae in China Seas

In this study, over 1051 yeast strains were obtained from the different marine environments (http://www.mccc.org.cn, 2 October 2009). After the identification of the marine-derived yeasts by using conventional and molecular methods, we found that only 17 strains were strains of *S. cerevisiae* (Tables 1 and 2 and Figure 1). The results in Table 1 show that most of them were isolated from skin, gills and contents of the gut and stomach of different species of marine fish in the East China Sea and Bohai Sea. Only the strain 2E00396 was obtained from the surface of *Laminaria japonica* collected from the Bohai Sea and the strain 2E00977 was obtained from the leaf surface of *Aegiceras* in the South China Sea. It was strange that no strains of *S. cerevisiae* were found in the marine fish and marine algae collected in the Yellow Sea, or the seawater, sediments and salterns obtained in all China seas.

**Table 1.** The strains of *S. cerevisiae* isolated from China seas.

| Strains | Strain Sources and Sampling Sites | Latitude and Longitude |
|---|---|---|
| 2E00396 | *Laminaria japonica*, Bohai Sea | 37°30′ N 122°10′ E |
| 2E00400 | Stomach of *Scomberomorus niphonius*, Bohai Sea | 37°30′ N 122°10′ E |
| 2E00498 | Gill of *Pseudosciaena crocea*, Bohai Sea | 37°30′ N 122°10′ E |
| 2E00550 | Gill of *Acanthopagrus schlegel*, Bohai Sea | 37°30′ N 122°09′ E |
| 2E00558 | Gill of *Pampus argenteus*, Bohai Sea | 37°30′ N 122°08′ E |
| 2E00561 | Skin of *Pampus argenteus*, Bohai Sea | 38°44′ N 118°49′ E |
| 2E00564 | Gill of *Gobio gobio*, Bohai Sea | 38°44′ N 118°49′ E |
| 2E00656 | Skin of *Pseudosciaena polyactis*, Bohai Sea | 38°44′ N 118°49′ E |
| 2E00723 | Stomach of *Acanthogobius flavimanus*, Bohai Sea | 37°30′ N 122°09′ E |
| 2E00724 | Digestive canals of *Paralichthys olivaceus*, Bohai Sea | 37°30′ N 122°08′ E |
| 2E00977 | Leaf surface of *Aegiceras*, South China sea | 21°07′ N 110°45′ E |
| 2E01006 | Gill of horsemackerel, East China Sea | 24°26′ N 118°02′ E |
| 2E01007 | Stomach of Dasyatis akaje, East China Sea | 24°26′ N 118°02′ E |
| 2E01008 | The skin of Goblet, East China Sea | 24°26′ N 118°02′ E |
| 2E01009 | The skin of *Muraenesox cinereus*, East China Sea | 24°26′ N 118°02′ E |
| 2E01010 | Intestine of *Stephanolepis cirrhife*, East China Sea | 24°26′ N 118°02′ E |
| 2E01011 | Stomach of *Saurida elongate*, East China Sea | 24°26′ N 118°02′ E |

**Table 2.** Results of carbohydrate fermentation tests and carbon assimilation of the marine yeasts.

| | 2E00396 | 2E00400 | 2E00498 | 2E00550 | 2E00558 | 2E00561 |
|---|---|---|---|---|---|---|
| **Fermentation** | | | | | | |
| Glucose | + | + | + | + | + | + |
| Maltose | + | − | + | + | + | − |
| Galactose | + | − | + | + | + | + |
| Sucrose | + | + | + | + | + | + |
| Lactose | − | − | − | − | − | − |
| Raffinose | + | + | + | + | + | + |
| **Assimilation** | | | | | | |
| Glucose | + | + | + | + | + | + |
| Maltose | + | + | + | + | + | + |
| Galactose | + | − | + | + | + | + |
| Sucrose | + | + | + | + | + | + |
| Lactose | − | − | − | − | − | − |
| Raffinose | + | + | + | + | + | + |
| Melibiose | − | − | − | − | − | − |
| Amidulin | − | − | − | − | − | − |
| Trehalose | + | − | + | + | + | + |

**Table 2.** *Cont.*

| | 2E00396 | 2E00400 | 2E00498 | 2E00550 | 2E00558 | 2E00561 |
|---|---|---|---|---|---|---|
| Cellobiose | − | +/W | − | − | − | − |
| D-arabinose | − | W/W | − | − | − | − |
| Xylose | − | W | − | − | − | − |
| L-arabinose | − | − | − | − | − | − |

| | 2E00564 | 2E00656 | 2E00723 | 2E00724 | 2E00977 | 2E001006 |
|---|---|---|---|---|---|---|
| **Fermentation** | | | | | | |
| Glucose | + | + | + | + | + | + |
| Maltose | + | − | + | + | + | − |
| Galactose | + | − | − | + | + | − |
| Sucrose | + | + | + | + | + | + |
| Lactose | − | − | − | − | − | − |
| Raffinose | + | + | + | + | + | + |
| **Assimilation** | | | | | | |
| Glucose | + | + | + | + | + | + |
| Maltose | + | + | + | + | + | + |
| Galactose | + | − | + | − | + | − |
| Sucrose | + | + | + | + | + | + |
| Lactose | − | − | − | − | − | − |
| Raffinose | + | + | + | + | + | + |
| Melibiose | − | − | − | − | − | − |
| Amidulin | − | − | − | − | − | − |
| Trehalose | + | − | + | + | + | + |
| Cellobiose | − | W | − | − | − | − |
| D-arabinose | − | − | − | − | − | − |
| Xylose | − | − | − | − | − | − |
| L-arabinose | − | − | − | − | − | − |

| KERRYPNX | 2E01007 | 2E01008 | 2E01009 | 2E01010 | 2E01011 | *S. cerevisiae* **ATCC 32703** |
|---|---|---|---|---|---|---|
| **Fermentation** | | | | | | |
| Glucose | + | + | + | + | + | + |
| Maltose | + | − | + | + | + | − |
| Galactose | + | − | − | + | + | − |
| Sucrose | + | + | + | + | + | + |
| Lactose | − | − | − | − | − | − |
| Raffinose | + | + | + | + | + | + |
| **Assimilation** | | | | | | |
| Glucose | + | + | + | + | + | + |
| Maltose | + | + | + | + | + | + |
| Galactose | + | − | + | − | + | − |
| Sucrose | + | + | + | + | + | + |
| Lactose | − | − | − | − | − | − |
| Raffinose | + | + | + | + | + | + |
| Melibiose | − | − | − | − | − | − |
| Amidulin | − | − | − | − | − | − |
| Trehalose | + | − | + | + | + | + |
| Cellobiose | − | W | − | − | − | − |
| D-arabinose | − | − | − | − | − | − |
| Xylose | − | − | − | − | − | − |
| L-arabinose | − | − | − | − | − | − |

+: positive result; −: negative result; w: weak.

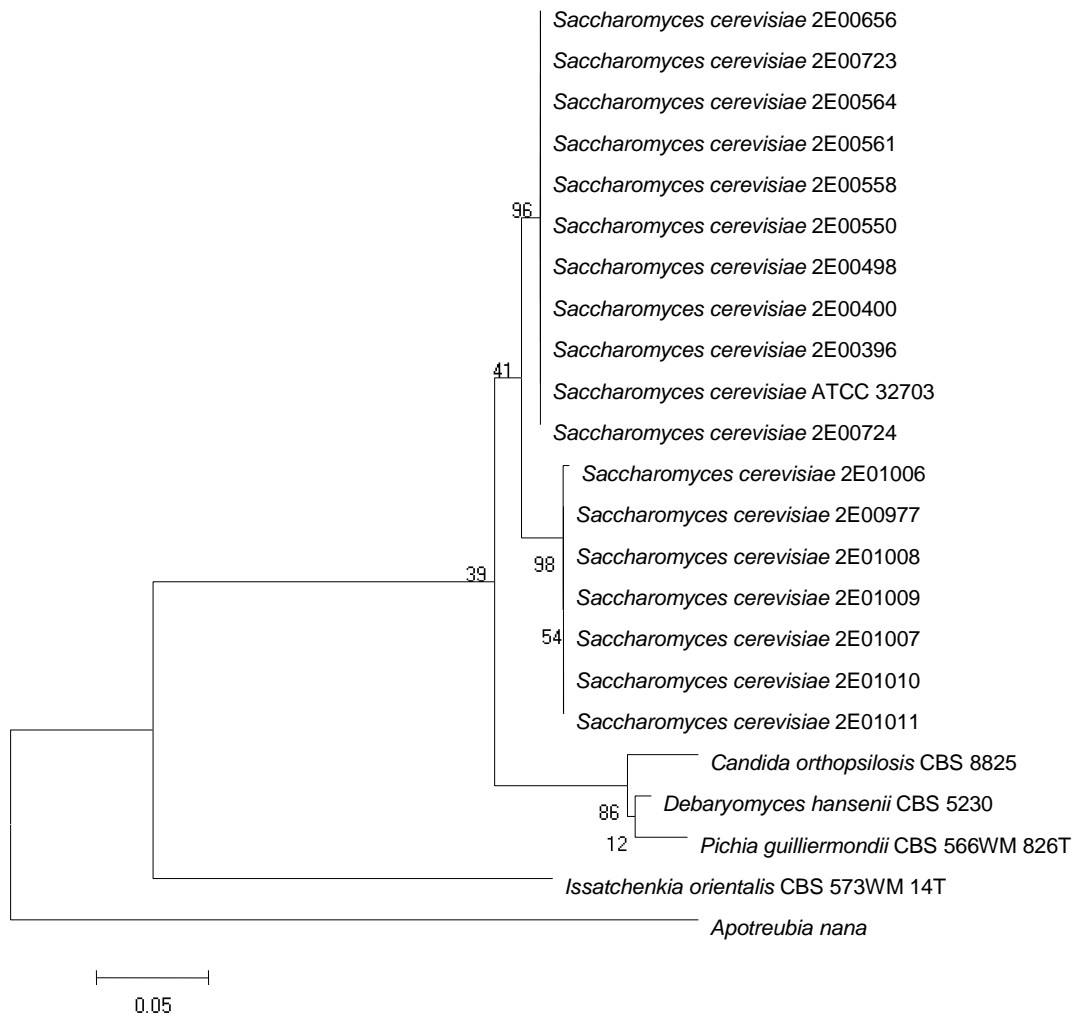

**Figure 1.** Phylogenetic tree of 17 strains of *S. cerevisiae* isolated in this study and 13 closest relatives based on a maximum parsimony analysis of D1/D2 26S rDNA sequences. Bootstrap values (1000 pseudoreplications) of ≥ 46%. All the strains shown are type strains and *Yarrowia lipolytica* was used as the out-group.

*3.2. Alcohol Production*

In this laboratory, *Saccharomyces* sp. W0 has been shown to produce over 15% ($v/v$) ethanol from sucrose and the hydrolysate of corn starch for a long time [21,22]. After determination of alcohol concentration in the fermented media by the 17 strains and *Saccharomyces* sp. W0, we found that only strains 2E00400, 2E00558, 2E00498, 2E00723, 2E00724 and W0 could produce a high concentration of ethanol (Figure 2) and any other strains of *S. cerevisiae* isolated in this study yielded less ethanol than them (data not shown). The results in Figure 2 show that the loss of weight by $CO_2$ liberation by the strains 2E00400, 2E00558, 2E00498, 2E00723 and 2E00724 during the fermentation, especially after 5 d of the fermentation, was lower than that by *Saccharomyces* sp. W0. The final alcohol concentrations produced by *Saccharomyces* sp. W0 were also higher than those produced by the strains 2E00400, 2E00558, 2E00498, 2E00723 and 2E00724. After the ANOVA analysis as described in the Materials and Methods, it was found that there were significant differences ($p = 0.015$) in the alcohol yield between *Saccharomyces* sp. W0 and all other yeast strains used in this study (data not shown). For example, *Saccharomyces* sp. W0 could produce 15.2% ($v/v$) ethanol while the strains 2E00400, 2E00558, 2E00498, 2E00723 and 2E00724 only produced 13.8% ($v/v$), 13.9% ($v/v$), 14.2% ($v/v$), 13.9% ($v/v$) and 14.0% ($v/v$) ethanol, respectively (data not shown). However, after the ANOVA analysis, it was found that there were no

significant differences ($p > 0.015$) in the ethanol yields between 2E00400, 2E00558, 2E00498, 2E00723 and 2E00724 (data not shown).

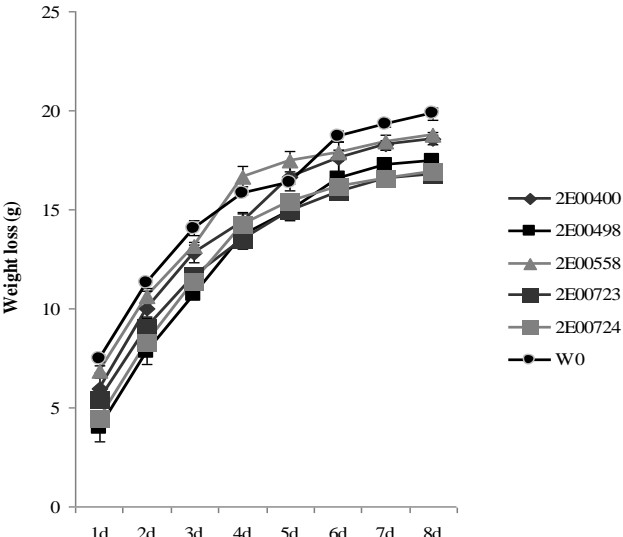

**Figure 2.** $CO_2$ liberation by fermentation of different yeast strains grown in the synthetic media containing 200.0 g L$^{-1}$ sucrose. Data are given as means $\pm$ SD, $n = 3$. All the experiments were done in triplicate, with the results averaged for each experiment.

*3.3. Alcohol Tolerance*

All the yeast strains tested in this study were grown in the YPD medium for 18 h and their cells were washed by centrifugation as described in the Materials and Methods. The washed cells were treated with an 18.0% ($v/v$) ethanol solution and cell survival was determined as described in the Materials and Methods. The results in Figure 3 indicated that greater cell survival of the yeast strain 2E00498 was observed than that of *Saccharomyces* sp. W0 within the first two hours of the ethanol shock treatment, although the latter could produce more ethanol than the former (Figure 2). For example, only 39% of the cells of *Saccharomyces* sp. W0 survived within the first two hours of the ethanol shock treatment while the strain 2E00498 maintained 51% survivors under the same conditions. This meant that the alcohol tolerance of the yeast strain 2E00498 isolated from the gills of *P. crocea* in the Bohai Sea (Table 1) was higher than that of *Saccharomyces* sp. W0, as supported by the ANOVA analysis ($p = 0.015$, differences were significant).

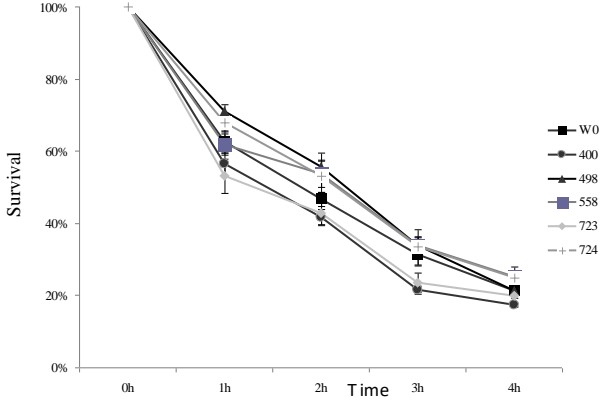

**Figure 3.** Cell survival (%) of different yeast strains during the high ethanol shock treatment. Data are given as means $\pm$ SD, $n = 3$.

It was found that that trehalose contents in the cells of many yeast strains isolated from marine environments, especially the yeast strain 2E00498, were also higher than those



in the cells of *Saccharomyces* sp. W0, as supported by the ANOVA analysis ($p = 0.015$, differences were significant) (data not shown).

## 4. Discussion

It has been well evidenced that different strains of *S. cerevisiae* can grow and survive in media rich in organic substances, such as the surfaces of the fruits of most species of plants and naturally fermented media [2]. Therefore, it is not surprising that most of the strains of *S. cerevisiae* obtained in this study were widely distributed in skin, gills and contents of the gut and stomach of different species of marine fish rich in organic substances (Table 1). Gatesoupe [8] reported that *Rhodotorula* sp. seemed relatively frequent in both marine and freshwater fish, and *Debaryomyces hansenii* has been found to be a main group in rainbow trout, but *S. cerevisiae* was not found in the marine samples. Other yeasts such as *Metschnikowia zobelii, Trichosporon cutaneum* and *Candida tropicalis* in marine fish, and *Candida* sp., *S. cerevisiae* and *Leucosporidium* sp. in rainbow trout, were also the dominant yeast populations. *S. cerevisiae* was obtained from marine invertebrates such as *Neoteredo reynei* (shipworm), *Anomalocardia brasiliana* (clam) and *Tagelus plebeius* (clam) in the southeast mangrove ecosystems in Brazil [23]. However, although the extreme diversity of yeast communities occurred in polluted estuary and mangrove ecosystems in subtropical marine, offshore and deep-sea environments, *S. cerevisiae* was not isolated in such environments, either [7]. Moreover, it is interesting to note from the results in Table 1 that all the strains of *S. cerevisiae* were isolated from only the samples of the Bohai Sea, the South China Sea and the East China Sea and no strain of *S. cerevisiae* was found in the samples of the Yellow Sea. In addition, it can also be clearly observed from the results in Figure 1 that all the yeast strains isolated from the Bohai Sea were clustered together into one group while 2E001006, 2E001007, 2E001008, 2E001009, 2E001010 and 2E001011 isolated from the East China Sea and only 2E000977 isolated from the South China Sea were clustered together into the same group. This may be due to the fact that the East China Sea and South China Sea may be suitable for growth of *S. cerevisiae* while the conditions and environments in the Yellow Sea are not suitable for the growth of *S. cerevisiae*. This was why no strain of *S. cerevisiae* was found in the samples of the Yellow Sea. This may imply that such geographical separation could produce a specific distribution of *S. cerevisiae* in marine environments. However, two marine yeast strains (S69 and S71) of *S. cerevisiae* were obtained from seawater from Plymouth, UK [24]. At the same time, two marine strains (S117 and S118) of *S. cerevisiae* were also found in rotten seaweed from Plymouth, UK [24]. *S. cerevisiae* C19 was isolated from Tokyo Bay by Obara et al. [25]. This means that the environmental conditions in the China Sea are different from those in any other sea in the world.

We found that only strains 2E00400, 2E00558, 2E00498, 2E00723 and 2E00724 could produce higher concentrations of ethanol than any other strains of *S. cerevisiae* obtained in this study (Figure 2). Although the five yeast strains isolated from marine environments could produce high concentrations of alcohol from sucrose, their ability to produce alcohol was lower than that of *Saccharomyces* sp. W0 which was isolated from fermented rice [11]. This means that *Saccharomyces* sp. W0 is indeed a good producer of ethanol. According to the sources of the five yeast strains isolated from the marine environments (Table 1), it can be clearly observed that all of them were obtained from different parts of marine fish collected from the Bohai Sea. However, all the strains of *S. cerevisiae* isolated from the East China Sea and the South China Sea produced lower concentrations of ethanol than the five yeast strains isolated from the Bohai Sea. Thus far, it is completely unknown why all the strains of *S. cerevisiae* isolated from the East China Sea and the South China Sea produce lower concentrations of ethanol than the five yeast strains isolated from the Bohai Sea. Therefore, more strains of *S. cerevisiae* need to be isolated from the seas to confirm this in the future.

It was found that the marine-derived yeast *S. cerevisiae* AZ65 isolated from seawater of the English Channel, Plymouth, UK could produce 11.4 g ethanol per 100 mL of fermented

medium using seawater–YPD medium and yielded 5.0 g ethanol per 100 mL of fermented medium using seawater–molasses medium [26]. Furthermore, the same marine-derived yeast *S. cerevisiae* AZ65 produced 97.0 g $L^{-1}$ ethanol from a glucose-based medium in a 15 L fermentor [27]. A marine-derived strain (JN387604) of *S. cerevisiae* that was obtained from mangrove soil on the southeast coast of India had higher bioethanol production (4.8 mg $L^{-1}$ of ethanol) from 6.84 mg $L^{-1}$ of sawdust than a terrestrial strain [28]. However, the amount of ethanol produced by the marine-derived strain (JN387604) was too low, so the technique was meaningless. The *S. cerevisiae* C19 strain isolated from Tokyo Bay yielded 123.0 g $L^{-1}$ and 88.0 g $L^{-1}$ of bioethanol from a concentrated paper shredder scrap hydrolysate and a mixture of seaweed hydrolysate (*Undaria pinnatifida*) and shredded paper, respectively [25,28]. This demonstrated that the marine-derived *S. cerevisiae* obtained in this study could produce a high ethanol concentration compared to any other marine-derived *S. cerevisiae* isolated from any other seas in the world (Figures 2 and 3).

The yeast cells of the strain 2E00498 also contained more trehalose than those of *Saccharomyces* sp. W0 (data not shown), suggesting that trehalose was involved in alcohol tolerance and some marine-derived yeast strains isolated from harsh environments indeed contained a high content of trehalose. This may partially explain why the alcohol tolerance of the yeast strain 2E00498 was higher than that of *Saccharomyces* sp. W0 (Figure 3). It has been well documented that trehalose in yeast cells can be involved in alcohol tolerance and other stress tolerances. This is due to the fact that trehalose in yeast cells can function as a highly efficient protectant for biological macromolecules such as proteins, enzymes and DNA, enhancing the resistance of the key cellular components against various harsh conditions such as high and cold temperatures, freezing, dehydration, various radiations, high osmotic pressure and high concentrations of ethanol and other solvents [29–34]. In our previous studies [35], the results showed that the high alcohol tolerance of the yeast cells of *Saccharomyces* sp. W0 was dependent of the trehalose content and trehalose in the yeast cells could even protect the integrity of the mitochondrial membrane and prevent the loss of mitochondrial DNA. However, in another study [10], it was also found that high ergosterol and phosphatidylinositol contents in the yeast cells may be responsible for the high alcohol tolerance and high ethanol yield of *Saccharomyces* sp. W0. It has also been reported that among different wine yeasts tested, *S. cerevisiae* var. *capensis* flor, with the highest ergosterol concentration in the plasma membrane of its cells, was found to be the most alcohol tolerant when it was shocked with 4.0% (*v/v*) alcohol [36]. As ethanol tolerance of the yeast strain 2E00498 was higher than that of *Saccharomyces* sp. W0, the ethanol tolerance of *Saccharomyces* sp. W0 could be improved by cell fusion or hybridization between the yeast strain 2E00498 and *Saccharomyces* sp. W0. This work is being carried out in this laboratory. Such recombinant strains of fusants or hybrids will be useful in the fermentation and biofuel industries.

## 5. Conclusions

In this study, it was found that most of the strains of *S. cerevisiae* in marine environments occurred in the guts and on the surface of marine fish and was not found in sea water and sediments. All the strains of *S. cerevisiae* isolated from the marine environments had a lower ability to produce ethanol than the highly alcohol-producing yeast *Saccharomyces* sp. W0. However, some of them had higher alcohol tolerance and a higher trehalose content than *Saccharomyces* sp. W0. Such marine-derived yeast strains with higher ethanol tolerance could be used to further improve the alcohol tolerance of *Saccharomyces* sp. W0 by cell fusion and hybridization. Fusants with both high alcohol production and high ethanol tolerance could be widely applied in various sectors of biotechnology.

**Author Contributions:** Conceptualization, Z.C., G.-L.L. and Z.-M.C.; methodology, B.-C.T.; Formal analysis, Z.H.; Resources, Z.-M.C.; data curation, B.-C.T.; writing—original draft preparation, Z.-M.C. and Z.H.; writing—review and editing, all authors; funding acquisition, Z.-M.C. All authors have read and agreed to the published version of the manuscript.

**Funding:** This work was supported by National Natural Science Foundation of China (Grant No. 31970058) and the Fundamental Research Funds for the Central Universities (Grant No. 31500029).

**Institutional Review Board Statement:** Not applicable.

**Informed Consent Statement:** Not applicable.

**Data Availability Statement:** The data presented in this study are available on request from the corresponding author.

**Acknowledgments:** This work was supported by the National Natural Science Foundation of China (Grant No. 31970058) and the Fundamental Research Funds for the Central Universities (Grant No. 31500029).

**Conflicts of Interest:** The authors declare no conflict of interest.

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
