# Peer review of "Occurrence and Distribution of Strains of Saccharomyces cerevisiae in China Seas"

_jmse, doi:10.3390/jmse9060590_

Round 1

Reviewer 1 Report

This manuscript “Occurrence and Distribution of Strains of Saccharomyces cerevisiae in Marine Environments" by Bai-Chuan Tian and colleagues identified several yeast strains from Saccharomyces cerevisiae collect in different marine environments in China Sea and their potential activity to produce ethanol. In general, the topic covered by the manuscript may have a potential interest to find new natural places where microbiological activity can be used for different purposes, namely in terms of food production. However, in my opinion, the manuscript has several points that must to be deeply improved before a possible publication. All manuscript is elaborated and presented in a confused way, without a correct and adequate planification. On the other hand, authors are not able to demonstrate the real relevance of the article nor in a clear way its objectives.

In the following comments, I’m showing several weaknesses of this work that must be change and improved. Thus, in summary I recommend that this manuscript must be resubmit.

I emphasize that it becomes very difficult to indicate the suggestions concisely because the document has a deficient annotation of the numbers of the lines.

Abstract

Line 3-4: Rewrite the sentence. Not clear or complete the sentence.

Line 8: “Saccharomyces sp. WO”, authors must to define in the abstract what’s Saccharomyces sp. WO means ? introduce only a few words. This points its relevant to the readers.

Line 5: Keywords: delete “Distribution; Occurrence” and change for “identification; ethanol production”.

Introduction

“beverages” ? however, wine and beer are beverages. Rewrite the sentence.

“wine-production regions”, not comprehensible in the context of the sentence. Rewrite.

At the end of the introduction, the authors must indicate the innovation of the work and the practical relevance of the research developed.

Material and methods

Saccharomyces sp. WO, this yeast strain was isolated in the laboratory or obtained by commercial purchase. This aspect needs to be clarified.

The samples were collected many years ago (2004, 2005 and 2008) and results are only now being published. Any explanation ?

“2.10. Ethanol Assay” Probably the GC conditions used were already described. Thus, introduce the reference that supports the analysis conditions used.

Results

“(Tables 1 and 2 and Figure 1)”. Only table 1 really shows the 17 different yeast strains for the first time. Table 2 and figure 1 show other aspects to be mentioned later. Thus, delete “.... and 2 Figure 1)”.

Table 2: In my opinion, this table must be restructured. Probably a single table in landscape format.

The results show in Table 1 are not clearly described in the text. Therefore, some sentences about the most important data showed in table 1 should be included.

Last sentence of page 8: “... environments (Tables 1 and 2, Figure 1) ....” It does not make sense to mention these tables and figures. They do not present data of the alcohol production by yeasts.

Page 9: “(data not shown) ...” In my opinion, these results must be show.

Figure 3: Statistical analysis results (ANOVA) must be introducing in the graphic, using letters and an adequate legend.

Page 9: line 6-9: This information it is not relevant for result section. It has already been mentioned in the material and methods. Delete in all similar situations throughout the article. The authors are repeating information already mentioned in the material and methods.

Page 9 “... than the former (Figure 2 and 3).” Figure 2 does not show results on ethanol production. It is suggested to delete.

Page 9: “The trehalose ..... and methods.” Not relevant sentence. Delete. The authors are repeating information already mentioned in the material and methods.

Table 3: Statistical analysis is missing (ANOVA). See previous comments for figure 3.

Page 12: Authors must to rewrite taking into consideration the following points: Introduce tentatively explanations for the results obtained; several sentences repeat aspects already showed in the results item; the reference to the map in figure 5 does not make sense. The interest of this figure is not perceived. It only reports on locations and does not present relevant information on the environmental conditions of the sample collection sites.

Page 13: For the first two paragraphs: Yes, according to the data from previous studies this is true. However, authors must to introduce a clear discussion and seek to present possible explanations for the results obtained or future research actions on this topic.

Page 13: “The results ..... shock treatment (Figure 4)”. The authors are repeating information already mentioned in the results item.

Page 13: However, authors must to introduce the potential relevance of this research and the results obtained.

Page 14: Conclusions: Authors are only describing the results obtained. They need to introduce the main global results, real relevance of the results obtained and also future research. All of these points are not well demonstrated.

Author Response

This manuscript “Occurrence and Distribution of Strains of Saccharomyces cerevisiae in Marine Environments" by Bai-Chuan Tian and colleagues identified several yeast strains from Saccharomyces cerevisiae collect in different marine environments in China Sea and their potential activity to produce ethanol. In general, the topic covered by the manuscript may have a potential interest to find new natural places where microbiological activity can be used for different purposes, namely in terms of food production. However, in my opinion, the manuscript has several points that must to be deeply improved before a possible publication. All manuscript is elaborated and presented in a confused way, without a correct and adequate planification. On the other hand, authors are not able to demonstrate the real relevance of the article nor in a clear way its objectives.

In the following comments, I’m showing several weaknesses of this work that must be change and improved. Thus, in summary I recommend that this manuscript must be resubmit.

I emphasize that it becomes very difficult to indicate the suggestions concisely because the document has a deficient annotation of the numbers of the lines.

Thank you very much for your kind comments and good suggestions. We must revise the manuscript very carefully.

Abstract

Line 3-4: Rewrite the sentence. Not clear or complete the sentence.

The sentence has been changed to “However, little has been known about the occurrence, distribution and roles of S. cerevisiae in marine environments.”

Line 8: “Saccharomyces sp. WO”, authors must to define in the abstract what’s Saccharomyces sp. WO means ? introduce only a few words. This points its relevant to the readers.

We have mentioned that Saccharomyces sp. W0 isolated from the fermented rice was a high alcohol-producing yeast in the Abstract

Line 5: Keywords: delete “Distribution; Occurrence” and change for “identification; ethanol production”.

“Distribution; Occurrence” have been deleted and were changed for “identification; ethanol production”.

Introduction

“beverages” ? however, wine and beer are beverages. Rewrite the sentence.

“and beverages,” has been deleted.

“wine-production regions”, not comprehensible in the context of the sentence. Rewrite.

This has been changed to “wine-producing sites”

At the end of the introduction, the authors must indicate the innovation of the work and the practical relevance of the research developed.

This has been changed to “The main aims of this study are to survey occurrence and distribution of S. cerevisiae from different marine environments in China in order to select the strains of S. cerevisiae with high alcohol endurance for their application in alcohol industries and basic research in biotechnology.”

Material and methods

Saccharomyces sp. WO, this yeast strain was isolated in the laboratory or obtained by commercial purchase. This aspect needs to be clarified.

Saccharomyces sp. W0, which was a both high alcohol-producing and high alcohol tolerant yeast strain, isolated from the fermented rice and had been utilized for high ethanol production this laboratory for over 20 years

The samples were collected many years ago (2004, 2005 and 2008) and results are only now being published. Any explanation ?

It is true that the samples were collected many years ago (2004, 2005 and 2008) and the yeast strains used in this study have been preserved at -80 °C for a long time. Only recently, we have been interested in doing the present research work because we have too many marine yeast strains.

“2.10. Ethanol Assay” Probably the GC conditions used were already described. Thus, introduce the reference that supports the analysis conditions used.

Ok, The GC conditions were described by Wang et al. [18].

Results

“(Tables 1 and 2 and Figure 1)”. Only table 1 really shows the 17 different yeast strains for the first time. Table 2 and figure 1 show other aspects to be mentioned later. Thus, delete “.... and 2 Figure 1)”.

I do not think so. The results in Table 1 can not show that the 17 different yeast strains belonged to S. cerevisiae. Only the data in Table 2 and Figure 1 can indicate this. Therefore, “.... and 2 Figure 1)” can not be deleted.

Table 2: In my opinion, this table must be restructured. Probably a single table in landscape format.

Table 2 will be restructured by editors.

The results show in Table 1 are not clearly described in the text. Therefore, some sentences about the most important data showed in table 1 should be included.

Last sentence of page 8: “... environments (Tables 1 and 2, Figure 1) ....” It does not make sense to mention these tables and figures. They do not present data of the alcohol production by yeasts.

The results in Table 1 show that most of them were isolated from skin, gill and contents of gut and stomach of different species of marine fish in the East China Sea and Bohai Sea. Only the strain 2E00396 was obtained from the surface of Laminaria japonica collected from Bohai Sea and the strain 2E00977 was got from leaf surface of Aegiceras, in the South China Sea. It was strange that no strains of S. cerevisiae were found in the marine fish and marine algae collected at the Yellow Sea, the seawater, sediments and salterns obtained in all China Seas.

Page 9: “(data not shown) ...” In my opinion, these results must be show.

These have been changed to “we found that only strains 2E00400, 2E00558, 2E00498, 2E00723, 2E00724 and W0 could produce high concentration of ethanol (Figures 2 and 3) and any other strains of S. cerevisiae isolated in this study yielded less ethanol than them (data not shown).”

Figure 3: Statistical analysis results (ANOVA) must be introducing in the graphic, using letters and an adequate legend.

Figure 3. Alcohol concentrations produced by different yeast strains grown in the synthetic media. Data are given as means ± SD, n=3. **There were significant differences (P = 0.015) of the alcohol yield between Saccharomyces sp. W0 and any other yeast strains.

And two *has been added.

Page 9: line 6-9: This information it is not relevant for result section. It has already been mentioned in the material and methods. Delete in all similar situations throughout the article. The authors are repeating information already mentioned in the material and methods.

“In order to evaluate the ability of all the strains of S. cerevisiae isolated from the marine environments (Tables 1 and 2, Figure 1) to produce alcohol in the synthetic medium as described in Materials and methods, Saccharomyces sp. W0 was used as control in this study.” Has been deleted.

Page 9 “... than the former (Figure 2 and 3).” Figure 2 does not show results on ethanol production. It is suggested to delete.

the loss of weight by CO2 liberation by the strains 2E00400, 2E00558, 2E00498, 2E00723 and 2E00724 during the fermentation, especially after 5 d of the fermentation was lower than that by Saccharomyces sp. W0.

I am sorry in Figure 2, “ethanol production” was not mentioned.

Page 9: “The trehalose ..... and methods.” Not relevant sentence. Delete. The authors are repeating information already mentioned in the material and methods.

“The trehalose contents in different yeast cells used in this study were measured as described in Materials and methods” has been deleted

Table 3: Statistical analysis is missing (ANOVA). See previous comments for figure 3.

**There were significant differences (P = 0.015) of the trehalose contents between the yeast strain 2E00498 and Saccharomyces sp. W0.

Page 12: Authors must to rewrite taking into consideration the following points: Introduce tentatively explanations for the results obtained; several sentences repeat aspects already showed in the results item; the reference to the map in figure 5 does not make sense. The interest of this figure is not perceived. It only reports on locations and does not present relevant information on the environmental conditions of the sample collection sites.

The Map in Figure 5 has been deleted and “the conditions and environments in Bohai Sea” has been deleted.

Page 13: For the first two paragraphs: Yes, according to the data from previous studies this is true. However, authors must to introduce a clear discussion and seek to present possible explanations for the results obtained or future research actions on this topic.

However, more strains of S. cerevisiae are needed to be isolated from the Seas to confirm this in the future.

Page 13: “The results ..... shock treatment (Figure 4)”. The authors are repeating information already mentioned in the results item.

“The results in Figures 2 and 3 revealed that Saccharomyces sp. W0 isolated from the fermented rice could produce more ethanol than the marine-derived strain 2E00498. But the strain 2E00498 had higher alcohol endurance than Saccharomyces sp. W0 during the ethanol shock treatment (Figure 4).” Has been deleted.

Page 13: However, authors must to introduce the potential relevance of this research and the results obtained.

As ethanol tolerance of the yeast strain 2E00498 was higher than that of Saccharomyces sp. W0, the ethanol tolerance of Saccharomyces sp. W0 could be improved by the cell fusant or hybridization between the yeast strain 2E00498 and Saccharomyces sp. W0. This work is being done in this laboratory. Such recombinant strains of fusant or hybridization will be useful in fermentation and biofuel industries.

Page 14: Conclusions: Authors are only describing the results obtained. They need to introduce the main global results, real relevance of the results obtained and also future research. All of these points are not well demonstrated.

Such marine-derived yeast strains with higher ethanol tolerance could be used to further improve alcohol endurance of Saccharomyces sp. W0 by the cell fusion and hybridization. The fusants with both high alcohol production and high ethanol tolerance will be widely applied to various sectors of biotechnology.

Reviewer 2 Report

This is an interesting research and contributes to development of food biotechnology and overall biotechnology. 

Minor revisions:

Line 34: CO2

Line 59: italics  Oncorhynchus mykiss (check the whole text)

Author Response

  1. Comments and Suggestions for Authors

This is an interesting research and contributes to development of food biotechnology and overall biotechnology. 

Thank you for your kind comments.

Minor revisions:

Line 34: CO2

CO2 has been changed to CO2

Reviewer 3 Report

Manuscript ID: jmse-1189717
Journal: Journal of Marine Science and Engineering
Title: Occurrence and Distribution of Strains of Saccharomyces cerevisiae in Marine Environments 
Authors: Bai-Chuan Tian, Guang-Lei Liu, Zhe Chi, Zhong Hu, Zhen-Ming Chi

The main aims of this study was to evaluate the occurrence and distribution of S. cerevisiae from different marine environments in China for selecting strains with high resilience to ethanol. Authors conclude that most of the isolates strains were not found in sea water or on sediments and all of them show low ability to produce ethanol compared to the terrestrial yeast strain. In a general appreciation the work has some novelty considering that marine habitats remain still unexplored for their microbial diversity. Therefore, this research to explore the diversity of culturable Saccharomyces cerevisae in such environment could be of interest for academia but it is of limited interest for industry of alcoholic beverages. The conclusions are very vague what reduces the interest to the readers.

Comments need revision are:

The title can be changed to “Occurrence and Distribution of Strains of Saccharomyces cerevisiae associated with marine fish in China Seas;

Abstract:  The abstract needs to be more focused and informative, the objectives and the relevance of the study, as well as the major results and conclusions must be clearly presented;

Introduction: This section must be focused on the important aspects that seem to be the knowledge of the geographical distribution of Saccharomyces in the Chinese Seas. Reviewer suggest authors contrast what has already been done on the subject, highlighting the relevance of the study. Long description about Saccharomyces cerevisiae/alcoholic fermentation, are aspects of common knowledge, such as lines 28-41, must be avoided. Only one statement shorter and concise it is recommendable;

Lines 61-63: Facts of importance are affirmed that should be cited results published in bibliography;

Lines 64-66: Assumptions not supported by results published in bibliography;

Material and Methods

There are several aspects that need to be introduced/clarified or explicated in Material and Methods (M&M) section. The methodologies should be described to an extent that makes understandable about what has been really done, and why these procedures were conducted.

Chemicals have to be given the name, where they have been purchased, company, city, and country. Describe in Material and Methods section the methodology used for carbohydrate fermentation and carbon assimilation tests (Results displayed in Table 2, lines 217-);

Gather together the subheading 2.1, 2.2 and 2.3 in the same item entitled “Marine environments, material sampled, and sampling”. Probably a picture/figure with geographic distribution of the sampled subareas is recommendable;

Open a new subheading for “Strains isolation, purification, and identification”, including 2.4 to 2.7. At the end of this item something can be stated about the strain WO of Saccharomyces cerevisiae, used as control as the representative terrestrial yeast.

Lines 147-157:  The methodology is not clear. Please rewrite all the methodology, clarify the culture medium used, the procedure of inoculation, yeast cell counting, etc. It is not clear how the yeasts grow with 200 gr of sucrose and 10g/L of ammonium sulphate;

Line 156-157: Clarify the concept and rephrase the statement “As the fermentation proceeded, additional sucrose was complemented to keep a suitable concentration of sucrose in the medium

Lines 159-174: The methodology needs to be clarified and to be very specific regarding the exact number of days yeast were cultured/maintained before the contact with the ethanol, the culture medium tested, the exact time of contact with ethanol in minutes; and the concentration of absolute ethanol used in the procedure;

Lines 168-170: combine the 2 sentences into a single one;

Lines 171-173: Accordingly, the results from duplicate plates were averaged, specify how the number of survivors of ethanol-treated strains are expressed. Probably as CFU per ml of ethanol-treated strains.

Lines 176-184- Ethanol assay and Lines 185-193 Threalose extraction”:  Methods that are already published should be briefly described and indicated by a reference.

All figures need improvement since they must be comprehensible without reference to the text.

Regarding Figure 2 caption I suggest “Fermentation profile of X yeast strains in Y medium, monitored by weight loss as an estimate of CO2 production”. Explain all symbols and abbreviations used in the caption, as well as the number of replicates that have been carries out. The same has to be done in Figure 4;

Figure 3 can be removed and the data described in text only. I also suggest that Table 3 can be deleted and data described in the text;  

A combined Results and Discussion section seems more appropriate in this case.  Anyway, results must be clear and concise. I think some tables must be avoided. Explore the more significant results of the work, and do not repeat them.

Author Response

  1. Comments and Suggestions for Authors

Manuscript ID: jmse-1189717
Journal: Journal of Marine Science and Engineering
Title: Occurrence and Distribution of Strains of Saccharomyces cerevisiae in Marine Environments 
Authors: Bai-Chuan Tian, Guang-Lei Liu, Zhe Chi, Zhong Hu, Zhen-Ming Chi

The main aims of this study was to evaluate the occurrence and distribution of S. cerevisiae from different marine environments in China for selecting strains with high resilience to ethanol. Authors conclude that most of the isolates strains were not found in sea water or on sediments and all of them show low ability to produce ethanol compared to the terrestrial yeast strain. In a general appreciation the work has some novelty considering that marine habitats remain still unexplored for their microbial diversity. Therefore, this research to explore the diversity of culturable Saccharomyces cerevisae in such environment could be of interest for academia but it is of limited interest for industry of alcoholic beverages. The conclusions are very vague what reduces the interest to the readers.

Thank you for kind comments.

Comments need revision are:

The title can be changed to “Occurrence and Distribution of Strains of Saccharomyces cerevisiae associated with marine fish in China Seas;

The title has been changed to “Occurrence and Distribution of Strains of Saccharomyces cerevisiae associated with marine fish in China Seas;

Abstract:  The abstract needs to be more focused and informative, the objectives and the relevance of the study, as well as the major results and conclusions must be clearly presented;

This has been done.

Introduction: This section must be focused on the important aspects that seem to be the knowledge of the geographical distribution of Saccharomyces in the Chinese Seas. Reviewer suggest authors contrast what has already been done on the subject, highlighting the relevance of the study. Long description about Saccharomyces cerevisiae/alcoholic fermentation, are aspects of common knowledge, such as lines 28-41, must be avoided. Only one statement shorter and concise it is recommendable;

“Therefore, it is widespread in wine, beer, sourdoughs, cider, sherry, cheese, sugar fermentation broth, indigenously fermented metabolites foods, soil, water, wine-producing sites and surface of fruits, stem, leaves of plants and so on” and “It is also used as the cell factories for the expression of the recombinant DNA and construction of new synthetic pathways in synthetic biology.” has been removed.

Lines 61-63: Facts of importance are affirmed that should be cited results published in bibliography;

One more reference has been added here.

Lines 64-66: Assumptions not supported by results published in bibliography;

One more reference has been added here.

Material and Methods

There are several aspects that need to be introduced/clarified or explicated in Material and Methods (M&M) section. The methodologies should be described to an extent that makes understandable about what has been really done, and why these procedures were conducted.

After the revision, all these have been clear.

Chemicals have to be given the name, where they have been purchased, company, city, and country.

All the chemicals used in this study were purchased from Sinopharm Chemical Reagents Co. Ltd, Shanghai, China.

Describe in Material and Methods section the methodology used for carbohydrate fermentation and carbon assimilation tests (Results displayed in Table 2, lines 217-);

We think “The fermentation and assimilation tests for each yeast strain were performed using the methods described by Kurtzman and Fell [12]” is enough because the tests are the common experiments at this moment.

Gather together the subheading 2.1, 2.2 and 2.3 in the same item entitled “Marine environments, material sampled, and sampling”. Probably a picture/figure with geographic distribution of the sampled subareas is recommendable;

We do not think so. it is not suitable to gather together the subheading 2.1, 2.2 and 2.3 in the same item entitled “Marine environments, material sampled, and sampling”. The results in Table 2 have given geographic distribution of the sampled subareas. So the results can not be repeated.

Open a new subheading for “Strains isolation, purification, and identification”, including 2.4 to 2.7. At the end of this item something can be stated about the strain WO of Saccharomyces cerevisiae, used as control as the representative terrestrial yeast.

It is not suitable to have such a new subheading for “Strains isolation, purification, and identification”, including 2.4 to 2.7. The strain WO of Saccharomyces cerevisiae, has been described in Subsection 2.1.

Lines 147-157:  The methodology is not clear. Please rewrite all the methodology, clarify the culture medium used, the procedure of inoculation, yeast cell counting, etc. It is not clear how the yeasts grow with 200 gr of sucrose and 10g/L of ammonium sulphate;

We think that 2.8. Alcohol Fermentation Tests has been described very clearly. The synthetic medium has been described in reference [11] and was not shown here.

It is very normal that the yeasts can grow well with 200 gr of sucrose and 10g/L of ammonium sulphate.

Line 156-157: Clarify the concept and rephrase the statement “As the fermentation proceeded, additional sucrose was complemented to keep a suitable concentration of sucrose in the medium

As the fermentation proceeded, additional sucrose (around 5.0 g per 100 mL of the fermentation medium) was complemented to keep a suitable concentration of sucrose in the medium.

Lines 159-174: The methodology needs to be clarified and to be very specific regarding the exact number of days yeast were cultured/maintained before the contact with the ethanol, the culture medium tested, the exact time of contact with ethanol in minutes; and the concentration of absolute ethanol used in the procedure;

We think that The subsection: 2.9. Ethanol Shock Treatment has been described very clearly.

Lines 168-170: combine the 2 sentences into a single one;

  1. Each cell on the plates was grown at 30°C for 48 h or 72 h and the colonies appeared on each plate were counted after 48 or 72 h incubation at 30°C.

Lines 171-173: Accordingly, the results from duplicate plates were averaged, specify how the number of survivors of ethanol-treated strains are expressed. Probably as CFU per ml of ethanol-treated strains.

Yes, you are right.

Lines 176-184- Ethanol assay and Lines 185-193 Threalose extraction”:  Methods that are already published should be briefly described and indicated by a reference.

The GC conditions were described by Wang et al. [18].

In fact, the Methods have been briefly described and indicated by a reference.

All figures need improvement since they must be comprehensible without reference to the text.

Regarding Figure 2 caption I suggest “Fermentation profile of X yeast strains in Y medium, monitored by weight loss as an estimate of CO2 production”. Explain all symbols and abbreviations used in the caption, as well as the number of replicates that have been carries out. The same has to be done in Figure 4;

All the experiments were done in triplicate, with the results averaged for each experiment.

Figure 3 can be removed and the data described in text only. I also suggest that Table 3 can be deleted and data described in the text;

I am sorry that the data in Figure 3 and Table 3 are important in this study and must be kept.

A combined Results and Discussion section seems more appropriate in this case.  Anyway, results must be clear and concise. I think some tables must be avoided. Explore the more significant results of the work, and do not repeat them.

The repeated contents in Discussion section have been removed.

Round 2

Reviewer 1 Report

The manuscript was really improved. Thus, this revised version its now very interesting and in my opinion able to be accepted. There are, however, only two suggestions for changes:

1 - Introduction (last sentence): Change "The main aims of this study ..." for "Thus, the main aims of this study ..."

2 - Material and methods (Last sentence of 2.10. Ethanol Assay): Change "The GC conditions were described ...." for "The GC conditions used were previously described by ...".

Author Response

Fig.3 and Table 3 have been removed.
